# Quantifying the role of ozone-caused damage to vegetation in the Earth system: A new parameterization scheme for photosynthetic and stomatal responses

Fang Li[1], Zhimin Zhou[2,1], Samuel Levis[3], Stephen Sitch[4], Felicity Hayes[5], Zhaozhong Feng[6], Peter B. Reich[7], Zhiyi Zhao[8, 9], Yanqing Zhou[10, 11, 9]

[1] International Center for Climate and Environment Sciences, Institute of Atmospheric Physics, Chinese Academy of Sciences, Beijing, 100029, China

[2] Aba Teachers University, Aba, 623002, China

[3] National Center for Atmospheric Research, Boulder, CO 80305, USA

[4] Faculty of Environment, Science and Economy, University of Exeter, Exeter, EX4 4RJ, UK

[5] UK Centre for Ecology & Hydrology, Bangor, Gwynedd LL57 2UW, UK

[6] Key Laboratory of Ecosystem Carbon Source and Sink, China Meteorological Administration (ECSS-CMA),School of Ecology and Applied Meteorology, Nanjing University of Information Science and Technology, Nanjing, 210044, China

[7] Department of Forest Resources, University of Minnesota, St Paul, MN 55108, USA

[8] State Key Laboratory of Numerical Modeling for Atmospheric Sciences and Geophysical Fluid Dynamics, Institute of Atmospheric Physics, Chinese Academy of Sciences, Beijing, 100029, China

[9] College of Earth and Planetary Sciences, University of Chinese Academy of Sciences, Beijing, 100049, China

[10] College of Ecology and Environment, Xinjiang University, Urumqi, 830046, China

[11] State Key Laboratory of Desert and Oasis Ecology, Xinjiang Institute of Ecology and Geography, Chinese Academy of Sciences, Urumqi, 830011, China

*Correspondence to*: Fang Li (lifang@mail.iap.ac.cn)

**Abstract.** Surface ozone ($O_3$) is the primary air pollutant threatening global vegetation. It typically reduces photosynthetic rate and stomatal conductance, leading to changes in carbon, water, and energy cycles, vegetation structure and composition, and climate. Several parameterization schemes have been developed to integrate the photosynthetic and stomatal responses to $O_3$ exposure in regional and global process-based models to simulate time- and space-varying $O_3$ plant damage and its cascading dynamic influence. However, these schemes are calibrated based on limited observations and often fail to reproduce the response relationships in observations, impeding accurate assessments of the role of $O_3$ plant damage in the Earth system. This study proposes a new parameterization scheme to utilize the



extensive observations from $O_3$ fumigation experiments to inform large-scale modeling. It is built on

4210 paired data points of photosynthetic and stomatal responses compiled from peer-reviewed literature, over six times larger than those employed in earlier schemes. Functions of phytotoxic $O_3$ dose (POD) are found to accurately reproduce the statistically significant linear or nonlinear relationships observed between POD and either relative leaf photosynthetic rate or relative stomatal conductance for needleleaf trees, broadleaf trees, shrubs, grasses, and crops. These eliminate the practice in earlier schemes of setting

response functions as constants and applying the response function from one vegetation type to another. It outperforms the old scheme in the Community Land Model (CLM) which skillfully reproduces the observed response for crop photosynthetic rate only. The nonlinear response functions we developed depict decreasing plant sensitivity with increases in POD, enabling models to implicitly capture the variability in plant ozone tolerance and the shift among plant species for both intra- and inter-PFT within

a vegetation type observed in the real world. Then, the new scheme is incorporated into the Community Earth System Model version 2.2 (CESM2.2), specifically its land component CLM5, to quantify the global impacts of present-day $O_3$ plant damage by comparing the simulations with and without $O_3$ plant damage. Results show that $O_3$ exposure reduces the global leaf photosynthetic rate by 8.5% and stomatal conductance by 7.4%, around half the estimates using the old scheme. Furthermore, the new scheme

improves global GPP simulations, decreasing RMSE by 11.1% relative to simulations without $O_3$ plant damage and by 11.7% compared to the old scheme. These results underscore the importance of including $O_3$ plant damage in large-scale process-based models and the effectiveness of the new scheme in global assessing and projecting the role of $O_3$ plant damage in the Earth system.

## 1   Introduction

Surface ozone ($O_3$) is a major air pollutant damaging natural and managed ecosystems worldwide (Reich 1987; Ainsworth et al., 2012; Gribacheva and Gecheva, 2019; Feng et al., 2021). It is mainly formed through complex photochemical reactions among nitrogen oxides ($NO_x$), volatile organic compounds (VOCs), methane ($CH_4$), and carbon monoxide (CO) (Chameides et al., 1988; Ainsworth et

al., 2012). The rapid pace of industrialization and urbanization has led to increased emissions of these precursors and climate warming, both contributing to a dramatic surge in global $O_3$ levels, with an increase of 32−71% since 1850 (Griffiths et al., 2021; Szopa et al., 2021; Tarasick et al., 2019). If

climate mitigation and pollutant control efforts remain weak, this alarming upward trend is projected to persist (Turnock et al., 2020; Griffiths et al., 2021).

Over the past decades, extensive $O_3$ fumigation experiments have been conducted across various plant species to quantify the harmful effects of ozone on plant physiological processes and to understand the underlying mechanisms (CLRTAP, 2017). They found that $O_3$ generally reduces leaf photosynthetic rate, which occurs mainly by decreasing the Rubisco enzyme content and activity and chlorophyll content in the chloroplast, altering chloroplast structure, impairing the electron transport chain, and decreasing

both mesophyll conductance and stomatal conductance (Lombardozzi et al., 2013; CLRTAP, 2017; Zhou et al., 2017; Xu et al., 2023). The $O_3$-induced decrease in stomatal conductance is primarily due to abscisic acid-triggered $Ca^{2+}$ entry into the guard cells (Pei et al., 2000; Wilkinson and Davies, 2010), inhibition of $K^+$ channels (Tran et al., 2013), disruption of signal transduction pathways (Wilkinson and Davies, 2010; Astier et al., 2017; Hassan et al., 2021), an increase in internal leaf $CO_2$ (Herbinger et al.,

2007), and, over the long term, damage to the stomatal apparatus (Kangasjärvi et al., 2005; Reich, 1987). The changes in leaf photosynthetic rate and stomatal conductance have cascading biological, physical, and chemical effects on the carbon, water, and energy fluxes of terrestrial ecosystems (Sitch et al., 2007; Lombardozzi et al., 2015; Unger et al., 2020; Ma et al., 2023). These effects can further slow plant growth, alter vegetation structure and composition, reduce crop yield and timber production (Mills et al., 2013;

Fuhrer et al., 2016; Tai et al., 2014, 2021; CLRTAP, 2017; Agathokleous et al., 2020; Sharps et al., 2022; Feng et al., 2022), as well as modify surface climate and atmospheric composition (Ainsworth et al., 2012; Sadiq et al., 2017; Arnold et al., 2018; Zhu et al., 2022).

     Three major parameterization schemes (Felzer et al., 2004; Sitch et al., 2007; Lombardozzi et al., 2015) have been proposed and used in process-based models for regional and global simulations of time-

and space-varying $O_3$ plant damage. These process-based models can be land surface models, Dynamic Global Vegetation Models (DGVMs), Global Gridded Crop Models (GGCMs), and Earth System Models (ESMs) (Tian et al., 2010; Clark et al., 2011; Lombardozzi et al., 2013; Val Martin et al., 2014; Lawrence et al., 2019; Emberson et al., 2022). To ensure inter-process harmonization and dynamic modeling of the downstream impacts resulting from the damage, these schemes consider $O_3$ effects on

photosynthetic rate and stomatal conductance, unlike Integrated Assessment Models (IAMs) (CLRTAP, 2017) which jump to estimate the influence of $O_3$ on crop yield and timber production directly and bypass $O_3$ impacts on all processes before harvest. In these schemes, the global photosynthetic and stomatal



responses are categorized by several vegetation types (needleleaf trees, broadleaf trees, grasses, shrubs, and crops) operating within a unified framework yet differentiated by parameters. The parameters are

obtained from synthetic analysis of observations to ensure robustness and representativeness, aligning with utilizing big data to inform big ecology concepts in microsystems research (Reichman et al., 2011; Soranno and Schimel, 2014) and the construction principles of large-scale process-based modeling (Bonan, 2019).

Felzer et al. (2004) developed a parameterization scheme based on the $O_3$ response relationships

established by Reich (1987) for needleleaf trees and crops and Ollinger et al. (1997, 2002) for broadleaf trees, and applied it to the Terrestrial Ecosystem Model (TEM). In this scheme, the photosynthetic response for the current month was a function of the product of stomatal conductance and AOT40 (accumulated ozone exposure in ppb-hr over an hourly concentration threshold of 40 ppb in daylight hours). To address the persistent damage resulting from past ozone exposure during the lifespan of a

leaf, Felzer et al. (2004) compounded the current month's ozone effect with that of the previous month. This scheme was later incorporated into the Dynamic Land Ecosystem Model (DLEM), with adjustments made to the time step shifting from a monthly to a daily resolution (Ren et al., 2007; Tian et al., 2010). However, it should be noted that the product of stomatal conductance and AOT40 lacks quantitative physical interpretation and fails to account for the impact of chronic ozone exposure at $O_3$

concentrations below 40 ppb.

$POD_Y$ (phytotoxic $O_3$ dose over a flux threshold of $Y$ nmol $O_3$ m$^{-2}$ s$^{-1}$) has become increasingly used in observational studies due to its clear physical interpretation (the cumulative stomatal uptake of ozone), comprehensive consideration of stomatal conductance, ozone concentration, and ozone exposure duration, as well as generally stronger correlation with ozone effects than AOT40 (Karlsson

et al., 2004; Pleijel et al., 2004, 2022). Correspondingly, Sitch et al. (2007, hereafter S07) proposed a scheme in which upper and lower thresholds of photosynthetic response to $O_3$ were a function of instantaneous ozone flux, and the photosynthetic response parameters were derived using an inverse method to fit the observed relationship of $POD_Y$ with crop yield (Pleijel et al., 2004) and needleleaf and broadleaf tree biomass (Karlsson et al., 2004). The scheme was developed in the land surface model

MOSES-TRIFFID (Met Office Surface Exchange Scheme-Top-down Representation of Interactive Foliage and Flora Including Dynamics) (Sitch et al., 2007), and subsequently used in JULES (Joint UK Land Environment Simulator, successor of MOSES-TRIFFID) (Clark et al., 2011; Oliver et al., 2018)


and the DGVM YIBs (Yale Interactive terrestrial Biosphere model) (Yue and Unger, 2015, developed

based on TRIFFID and CASA). However, S07 has several limitations. First, due to a lack of

observational data collection and analyses, S07 applied crop and broadleaf tree response functions to

grasses and shrubs, respectively. Second, the photosynthetic response parameters derived through the

inverse method based on an observed relationship of $POD_Y$ with yield or biomass rather than with

photosynthesis directly are likely biased, influenced by uncertainties in simulating the processes (e.g.,

respiration, allocation, and phenology) and environmental variables such as soil moisture between

photosynthesis and harvest. Third, because the estimated parameters are model-dependent, applying

them directly to non-TRIFFID models may reduce the accuracy of $O_3$ damage simulations. Fourth, S07

models the upper and lower response thresholds, rather than the optimal representation as other

processes adopted. Lastly, S07 assumed the response function being the same for photosynthetic rate

and stomatal conductance, contradicting the increasing observations that chronic ozone exposure

decouples stomatal conductance and photosynthetic rate (Tjoekler et al., 1995; Wittig et al., 2007;

Lombardozzi et al., 2012; Kinose et al., 2020).

       To address these limitations, Lombardozzi et al. (2013, 2015) developed a scheme (hereafter L15)

that adopted different functions of $POD_Y$ for photosynthetic and stomatal response, based on 652 paired

data points of $POD_Y$ and relative photosynthetic rate/stomatal conductance compiled from the peer-

reviewed literature. The scheme was implemented in the land surface model CLM (Community Land

Model), the land component of the Community Earth system model (CESM) (Lawrence et al., 2019).

However, since the response function was assumed to be linear, L15 found a skillful (regression

statistically significant at the 0.05 level) response function for only crop photosynthetic rate and

temperate evergreen tree stomatal conductance. For other vegetation types, a constant (intercept of the

linear regression) was employed (see Appendix), resulting in a fixed simulated ozone effect regardless

of $POD_Y$ change. Furthermore, similar to S07, L15 applied the response functions of trees and crops to

shrubs and grasses due to no observations collected and no significant linear fitting found, respectively.

       In this study, we propose a novel parameterization scheme, in which the photosynthetic and

stomatal response functions are built upon 4210 paired data points collected from experimental

measurements reported in peer-reviewed literature. The sample size is over six times that of L15 and 23

times that of S07. Furthermore, we remove the linear assumption employed in prior schemes, and

identify 2-parameter linear or nonlinear functions of $POD_Y$ to capture the statistically significant





response relationship in observations for broadleaf trees, needleleaf trees, shrubs, grasses, and crops,

respectively. We then apply this scheme to CESM2.2's land component CLM5 to quantify the global

impact of present-day ozone exposure on the total, spatial distribution, and seasonality of leaf

photosynthetic rate and stomatal conductance. In addition, given the close relationship of gross primary

productivity (GPP) of terrestrial ecosystems with leaf photosynthesis and stomatal conductance, and

the availability of global GPP benchmark data, we evaluate global GPP simulations with and without

ozone stress and with different parameterization schemes.


## 2   Materials and methods

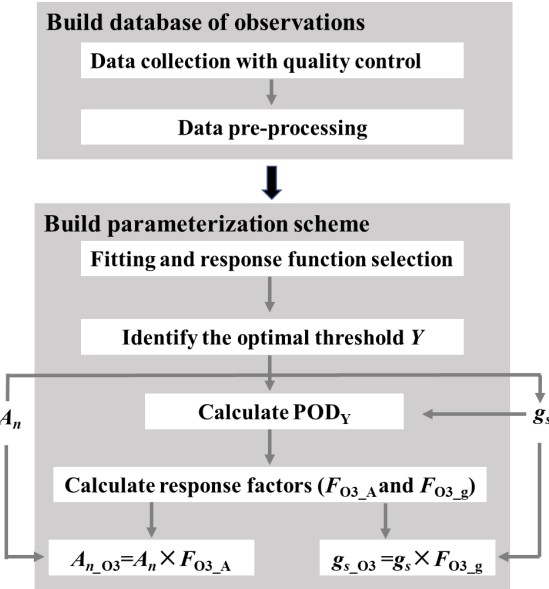

**Figure 1.** Flowchart illustrating the construction of $O_3$ plant damage parameterization scheme. $POD_Y$

(phytotoxic ozone dose over an ozone flux threshold of $Y$) represents the cumulative leaf stomatal

uptake of $O_3$; $A_n$ and $g_s$ are net photosynthetic rate and stomatal conductance without ozone plant

damage, respectively, while $A_{n\_O3}$ and $g_{s\_O3}$ are those modified by $O_3$ plant damage; $F_{O3\_A}$ and $F_{O3\_g}$ are

photosynthetic and stomatal response factors, respectively.

The parameterization scheme construction involves two steps (Fig. 1). First, we establish a database via

data collection with quality control and data preprocessing. Second, using the preprocessed data, we



construct the parameterization scheme through regression analysis, response function selection, identification of the optimal threshold $Y$, and incorporation of photosynthetic and stomatal response functions into regional and global process-based models. After the scheme construction, we apply it to the CESM2.2's land component CLM5 for quantifying the global impact of $O_3$ plant damage.

## 2.1 Construction of observational database

### 2.1.1 Data collection with quality control

A database of $O_3$ effects on leaf photosynthetic rate and stomatal conductance is compiled by conducting a survey of the peer-reviewed literature using keyword searches in the Web of Science, Springer Nature, and China National Knowledge Infrastructure. A total of 298 articles published from January 1970 to December 2022 have been identified to report experimental measurements on the $O_3$ effect. Measurements within an article are considered independent data points if they were made for different species, distinct genotypes within a species, different ozone treatments, or on different dates, consistent with the approach taken by Wittig et al. (2007) and Lombardozzi et al. (2013). Otherwise, they are treated as a sample of one data point, and the sample mean is used as a data point for analysis.

**Table 1.** Overview of experimental data collected from peer-reviewed literature about $O_3$ effects on photosynthetic rate and stomatal conductance. The numbers in parentheses are the number of articles, species, and data points within each category.

| Category | Categorical level | | | | |
|---|---|---|---|---|---|
| Plant type | BT (81, 87, 3902) | NT (21, 13, 669) | Crop (52, 117, 2293) | Grass (9, 18, 266) | Shrub (4, 4, 256) |
| Plant age (year) | <1 (63, 135, 2733) | 1 to 5 (57, 54, 2735) | >5 (12, 8, 200) | N/A (40, 65, 1718) | |
| Control air | Charcoal filtered (86, 145, 4399) | Ambient (48, 71, 1927) | Non-Filtered (6, 7, 198) | N/A (23, 39, 862) | |
| Exposure system | Growth chamber (41, 57, 1738) | Free-Air enrichment (28, 33, 1583) | Open top chamber (75, 139, 3240) | Greenhouse (17, 30, 756) | Branch chamber (2, 2, 69) |
| Rooting environ. | Pot (116, 183, 5178) | Ground (26, 36, 1083) | N/A (19, 33, 1125) | | |
| Response variable | Photosynthesis (140, 211, 3496) | Stomatal conductance (158, 236, 3890) | | | |

Data quality control is then carried out. Data points are excluded (1) if $POD_Y$ or variables for calculating $POD_Y$ (see Eq. 1) cannot be extracted; that is, only data categorized as high and medium



confidence defined by Lombardozzi et al. (2013) are considered in our study; (2) if either

photosynthetic rate or stomatal conductance, including their units, cannot be extracted or are

unreasonable (incorrect units or numerical deviation exceeds an order of magnitude); (3) if the data are

previously or more completely reported in another article; (4) if the photosynthetic rate is not reported

in conjunction with stomatal conductance; (5) if other environmental interactions are included so that

the effect of only $O_3$ is unclear; or (6) if experiments are conducted for fewer than 7 days and thus not

representative of chronic exposure. Following these criteria, data are collected from a total of 159

articles (see Supplements), representing 238 species and providing 3496 data points for photosynthetic

rate and 3890 data points for stomatal conductance (Table 1).

Stomatal conductance and photosynthetic rate or their relative values to those without ozone

stress, as well as $POD_Y$ or variables to calculate it in the control and elevated $O_3$ treatments are

collected from tables, figures, and text in the articles and compiled into a database. In cases where data

are presented graphically, PlotDigitizer v3 is employed for data extraction. When $POD_Y$ needs to be

calculated, but the light exposure of field experiments is not reported, sunlight duration is obtained

from the website https://richurimo.bmcx.com/9.61__jw__45.69__time__2013_09__richurimo/ by

entering the longitude, latitude, and date of the experiments. Additional information such as location,

vegetation type, plant species, plant age, type of control air, $O_3$ exposure system conditions, rooting

environment, sample size, sample standard deviation (SD) or standard error (SE), and reference are

also recorded for each data point, and summarized in Table 1.

### 2.1.2 Data pre-processing

For literature that does not provide $POD_Y$ (mmol $O_3$ m$^{-2}$), we calculate it for various candidates of $O_3$

flux threshold $Y$ (nmol $O_3$ m$^{-2}$ s$^{-1}$), using data from the literature on $O_3$ concentration at the leaf surface

($[O_3]_{ls}$, ppb), leaf stomatal conductance ($g_s$, mol $H_2O$ m$^{-2}$ s$^{-1}$), and the number of hours of plant

exposure to $O_3$ and light ($h$, hour), as:

$$POD_Y = \max([O_3]_{ls} \frac{g_s}{k_{O3}} - Y, 0) \times h \times 3600 \times 10^{-6}, \tag{1}$$

where $k_{O3}$=1.51 (=1/0.663) (mol $H_2O$ (mol $O_3$)$^{-1}$) is the ratio of leaf resistance for $O_3$ to that for water

vapor. Eq. (1) combines Eqs. (1) and (2) used in Lombardozzi et al. (2013) for preprocessing

observations, but with three modifications: $Y$ is not arbitrarily set to zero; a typo is corrected that $k_{O3}$

was incorrectly multiplied in Eq. (2) when it should have been divided; $k_{O3}$ value is updated based on



Massman et al. (1998) and CLRTAP (2017), instead of 1.67 used in Lombardozzi et al. (2013). The $Y$ candidates considered in this study cover all the values used in earlier studies, including 0.5, 0.8, 1, 1.6, 2, 3, 4, 5, and 6. Specifically, L15 used 0.8 for all plant types; S07 assigned 1.6 to needleleaf and broadleaf trees and shrubs, and 5 to crops and grasses; CLRTAP et al. (2017) applied 1 for natural

plants and 6 for crops, followed by Oliver et al. (2018) and Ma et al. (2023).

To achieve comparability of the $O_3$ effect across different experiments, species, control air types, and dates within a given vegetation type, we need to calculate the relative photosynthetic rate and relative stomatal conductance to the values without ozone stress if the literature does not report them. Similar to Karlsson et al. (2004), Pleijel et al. (2004), and Hayes et al. (2021), for pairs of control and

$O_3$-elevated experimental measurements that differ solely in ozone concentration, we begin by performing a simple linear regression, using photosynthetic rate or stomatal conductance as the dependent variable and $POD_Y$ as the independent variable. The regression enables us to obtain the intercept representing the photosynthetic rate or stomatal conductance at $POD_Y=0$. Next, we derive the relative values through dividing the photosynthetic rate or stomatal conductance by the intercept. Then,

we conduct linear regression using the relative values and corresponding $POD_Y$ for individual plant species in a study, and data with intercept falling outside the range of 0.9 to 1.1 are removed based on Hayes et al. (2021) and the LRTAP convention to ensure a usable response relationship. Finally, we exclude the paired data points at $POD_Y=0$. For literature that reports the relative photosynthetic rate or relative stomatal conductance in units of %, we convert it to a unitless fraction via dividing it by 100.

Through the above data pre-processing, we obtain the paired data points of $POD_Y$ and relative photosynthetic rate (or relative stomatal conductance), including 567−943 (or 486−1281) for broadleaf trees, 2−217 (or 3−232) for needleleaf trees, 0−153 (or 0−149) for shrubs, 20−40 (or 42−78) for grasses, and 380−605 (or 418−691) for crops (Tables S1−2). For a specific vegetation category, the values represent the ranges of the number of paired data points across different ozone flux thresholds $Y$. A higher

threshold $Y$ often results in more $POD_Y$ values equaling zero (Eq. 1), so more paired data points at $POD_Y=0$ are excluded during pre-processing to ensure a usable response relationship (refer to the last pre-processing step). The number of paired data points clearly varies with the threshold $Y$ for a specific vegetation type.

**2.2 Construction of the parameterization scheme**



### 2.2.1 Regression analysis and selection of response function


We use 2-parameter linear ( $y = ax + b$ ) and nonlinear ( $y = \mathrm{f}(x)$ ) regression functions to fit the pre-processed data, where $y$ is the relative photosynthetic rate or relative stomatal conductance and $x$ is $\mathrm{POD_Y}$, and f denotes a nonlinear function. For nonlinear regression, we consider five commonly used linearizable function forms: (i) logarithm function $y = a\ln(x) + b$, (ii) power function $y = bx^a$, (iii)


exponential function $y = b\mathrm{e}^{ax}$, (iv) hyperbolic tangent function $y = a\tanh(x) + b$, and (v) reciprocal function $y = \dfrac{a}{x} + b$. When parameter $a < 0$, the first four nonlinear functions and the linear function all imply a decrease in $y$ as $x$ increases. We use the least squares principle to estimate the regression coefficients and F-test to test the statistical significance of regression (Huang et al., 2016).

For each vegetation type and each ozone flux threshold $Y$, the sample is the same, so we compare


the residual sum of squares (RSS), which is the sum of the squared distances between observed versus predicted values, across different function forms. The function with the lowest RSS is identified as the optimal function.

As shown in Fig. 2, the linear function is typically optimal for needleleaf tree and grass photosynthetic response as well as crop stomatal response. The exponential function is often optimal


for broadleaf tree and crop photosynthetic response, while the logarithm for broadleaf tree stomatal response and for grass when Y is small.

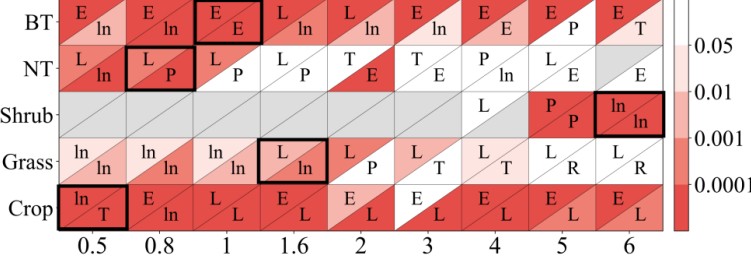

**Figure 2.** P-value from regression analysis using a linear or nonlinear function corresponding to different


ozone flux thresholds $Y$ (0.5, 0.8, 1, 1.6, 2, 3, 4, 5, 6) in $\mathrm{POD_Y}$ for photosynthetic (upper triangle) and stomatal (lower triangle) responses across different vegetation types: broadleaf trees (BT), needleleaf trees (NT), crops, grasses, and shrubs. A lower P-value (deeper red) indicates a regression with greater statistical significance and higher accuracy, and a regression with P<0.05 is considered statistically significant. The letters within the triangles denote the optimal function forms for a given $Y$ and vegetation





type determined by the smallest residual sum of squares (RSS): linear function (L), logarithm function (ln), exponential function (E), hyperbolic tangent function (T), power function (P), and reciprocal function (R). Boxes with the optimal $Y$ (required to be statistically significant for both photosynthetic and stomatal responses and have the highest significance) are outlined using a black frame. Triangles in gray indicate the number of observations less than 3.


### 2.2.2 Selection of ozone flux threshold $Y$

The optimal threshold $Y$ for each vegetation type is selected based on two criteria: (i) the P-values of the optimal regression functions for both the photosynthetic rate and stomatal conductance are less than 0.05 (i.e., regression is statistically significant) and (ii) the sum of the P-values for the $Y$ is smallest

(i.e., highest statistical significance).

Because a smaller sample size leads to fewer degrees of freedom, a higher coefficient of determination ($R^2$) is associated with a statistically significant regression model that is superior to a random prediction model. In our study, sample size obviously varies with threshold $Y$ (Tables S1−2 and Sect. 2.1.2), and comparison $R^2$ among different $Y$ fails to account for the effect of sample size.

Therefore, we use the P-value as the criterion, rather than $R^2$.

Consequently, the optimal threshold $Y$ is identified as 1 for broadleaf trees, 0.8 for needleleaf trees, 6 for shrubs, 1.6 for grasses, and 0.5 for crops (Fig. 2). The number of paired data points corresponding to the selected $Y$ is 2183 (=902 for photosynthetic response +1281 for stomatal response) for broadleaf trees, 326 (=140+186) for needleleaf trees, 302 (=153+149) for shrubs, 103 (=

27+76) for grasses, and 1296 (=605+691) for crops, totally 4210 (Table 2).

**Table 2.** The number of paired data points used to generate response functions of photosynthetic rate and stomatal conductance for the new parameterization scheme, L15 (Lombardozzi et al., 2015), and S07 (Sitch et al., 2007).

| Veg. type | New | L15 | S07 |
|---|---|---|---|
| BT | 2183 | 266 | 45 |
| NT | 326 | 100 | 51 |
| Shrub | 302 | 0 | 0 |
| Grass | 103 | 16 | 0 |
| Crop | 1296 | 270 | 80 |
| Total | 4210 | 652 | 176 |





### 2.3 Application for global simulations

#### 2.3.1 Model platform

We test the new parameterization scheme using the CESM. CESM is a widely utilized Earth system model that enables the simulation of global atmosphere, ocean, land, and sea ice, along with their interactions (Danabasoglu et al., 2020). It is developed by the CESM community and hosted at the National Center for Atmospheric Research (NCAR). For our study, we adopt the latest version CESM2.2, which incorporates CLM5 as its land component (Lawrence et al., 2019).

CLM5 uses the Farquhar-Collatz model for photosynthesis and the Medlyn model for stomatal conductance. When calculating photosynthesis and stomatal conductance, it distinguishes between sunlit and shaded leaves, in which sunlit leaves absorb both direct and diffuse solar radiation, while shaded leaves only receive diffuse radiation. The L15 scheme (see Appendix A) is included in CLM5 as an option to account for ozone damage to vegetation, but it is inactive in default simulations. Like L15, we calculate the $O_3$ uptake and its influence on the photosynthetic rate and stomatal conductance for sunlit and shaded leaves separately.

#### 2.3.2 Experimental design

We use the component set I2000Clm50Sp of CESM2.2 for present-day land offline simulations, similar to I2000Clm45Sp used in Lombardozzi et al. (2015). In this component set, CLM5 includes one bare soil PFT and 16 vegetated PFTs (three needleleaf tree PFTs, five broadleaf tree PFTs, three shrub PFTs, three grass PFTs, and two crop PFTs). The component set uses prescribed present-day vegetation distribution and structure and keeps the biogeochemical module inactive, so the impacts of $O_3$ plant damage on them and their feedback are not considered. It is acceptable because this study aims to quantify the direct photosynthetic and stomatal responses to $O_3$ exposure.

Three experiments are performed: $O_3$_New, $O_3$_OFF, and $O_3$_L15. The three simulations are identical except for the application of the new scheme proposed in this study, no ozone plant damage, and the L15 scheme, respectively. We quantify the global impacts of $O_3$ on leaf photosynthetic rate and stomatal conductance by calculating the difference between $O_3$_New and $O_3$_OFF and assess the impact of the different schemes by calculating the difference between $O_3$_New and $O_3$_L15.

All simulations are conducted for 30 years with 2005–2014 atmospheric forcing and surface ozone





concentration cycling 3 times. They are initialized from an equilibrium (spun-up) state of CLM5

default present-day simulations provided by CESM2.2, similar to O$_3$_OFF but employing a different

length of atmospheric forcing. The last 20 years of the simulations are analyzed, and the first 10-years

of the simulations are discarded as spin-up. The simulations run at a spatial resolution of 1.9° latitude

by 2.5° longitude, with a model time step of 30 minutes.

### 2.3.3 Input data

The Global Soil Wetness Project (GSWP3.1) provides a 3-hourly 0.5° global atmospheric reanalysis

dataset for 2005–2014, which serves as the default atmospheric forcing for CLM5. It includes variables

such as surface air temperature, wind speed, specific humidity, air pressure, insolation, and

precipitation. The input data of the prescribed present-day vegetation distribution and structure (LAI

and canopy height) are based on MODIS satellite observations and have no interannual variability. The

above-mentioned forcing and initial data, as well as atmospheric $CO_2$ concentration and nitrogen and

aerosol deposition for the year 2000, are provided with CESM2.2.

In our study, 2005–2014 time-varying surface air ozone concentration in ppb (i.e., volume mixing

ratio, VMR) is derived based on the 3-hourly 0.75° surface ozone mass mixing ratio (MMR, kg kg$^{-1}$)

from CAMS global reanalysis EAC4 (ECMWF Atmospheric Composition Reanalysis 4, Inness et al.,

2019) through multiplying MMR by $28.9644/47.9982 \times 10^9$ (Guisti, 2019). It is better than a global

constant ozone concentration set in CLM5 and time-step data from linear interpolation of monthly

ozone concentration used in the ongoing CLM development version. The ozone concentration in ppb

could convert to that in unit of nmol m$^{-3}$ used in Eq. (7) through multiplying by $P_{atm}/(\Theta_{atm} \times R) \times 1000$,

where $P_{atm}$, $\Theta_{atm}$, and R are atmospheric pressure (Pa), atmospheric potential temperature (K), and

universal gas constant (J K$^{-1}$ kmol$^{-1}$), respectively. In CESM coupled land-atmosphere simulations (not

performed here), ozone concentration can be simulated by the atmospheric model and transferred to the

land model.

### 2.3.4 Benchmark data

The FLUXCOM product is used as benchmark data to assess 2005–2014 averaged global GPP

simulations. The 0.5° daily FLUXCOM RS + METEO GPP product is derived by using machine

learning to integrate FLUXNET site-level observations, satellite remote sensing, and meteorological





data (Jung et al., 2020). It is commonly used to evaluate GPP simulations of regional and global

process-based models.

## 3    Parameterization scheme

Following the processes detailed in Sects. 2.1 and 2.2, photosynthetic and stomatal response functions

are generated (Figs. 3−4). The response factors of photosynthetic rate to $O_3$ ($F_{O3\_A}$, unitless) for

broadleaf trees (BT), needleleaf trees (NT), shrubs, grasses, and crops are given as

$$
F_{O3\_A} = \begin{cases}
0.943e^{-0.0085POD_1} & \text{BT} \\
1.005 - 0.0064POD_{0.8} & \text{NT} \\
1.000 - 0.074\ln(POD_6) & \text{Shrub} \\
0.997 - 0.016POD_{1.6} & \text{Grass} \\
0.909 - 0.028\ln(POD_{0.5}) & \text{Crop}
\end{cases}
\tag{2}
$$

and the response factors of stomatal conductance to $O_3$ ($F_{O3\_g}$, unitless) are

$$
F_{O3\_g} = \begin{cases}
0.943e^{-0.0058POD_1} & \text{BT} \\
0.965POD_{0.8}^{-0.041} & \text{NT} \\
0.991 - 0.060\ln(POD_6) & \text{Shrub} \\
0.989 - 0.045\ln(POD_{1.6}) & \text{Grass} \\
1.005 - 0.169\tanh(POD_{0.5}) & \text{Crop}
\end{cases}
\tag{3}
$$

As shown in Figs. 3−4, the regression is statistically significant for all vegetation types, so there is no

need to use a function from one vegetation type for another. This differs from earlier parameterization

schemes that employed substitution when regressions were not statistically significant or observations

were not available or collected for a specific vegetation type. When we evaluate the L15 scheme using

our expanded collected dataset, we find regression functions of L15 with $POD_{0.8}$ as the independent

variable are statistically significant for only crop photosynthetic rate. Even for the crop photosynthetic

rate, our scheme improves the fitting skill (quantified by $R^2$) by 8.1% (Table 3). As in L15, the response

factors are required to range from 0 to 1 to avoid unwanted outcomes in any scenario when used in

models.



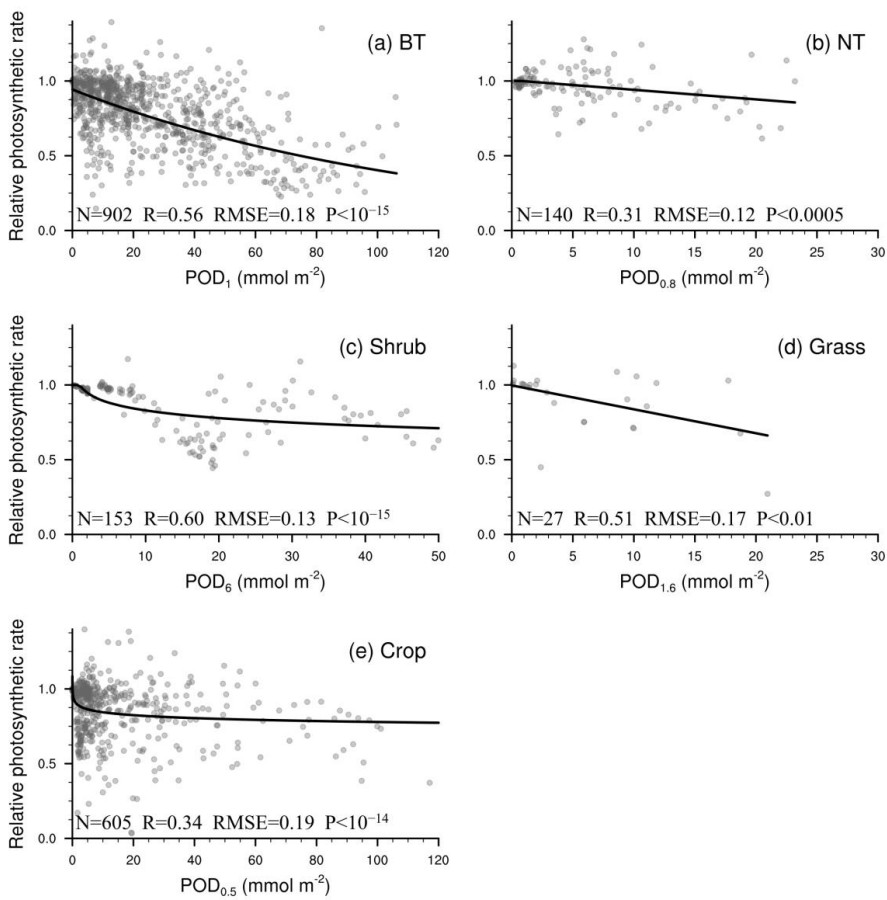

**Figure 3.** Relationship between $POD_Y$ and relative photosynthetic rate from experimental measurements (dots). The line of best fit (line) represents the photosynthetic response function ($F_{O3\_A}$) used in our parameterization scheme. Sample size of measurements (N), correlation coefficient (R), root mean square error (RMSE) between measurements versus predicted values, and P-value of regression (P) are also shown. When $P < 0.05$, the regression analysis is considered statistically significant. A smaller P-value indicates that the regression analysis has a stronger statistical significance and higher skill than random prediction.

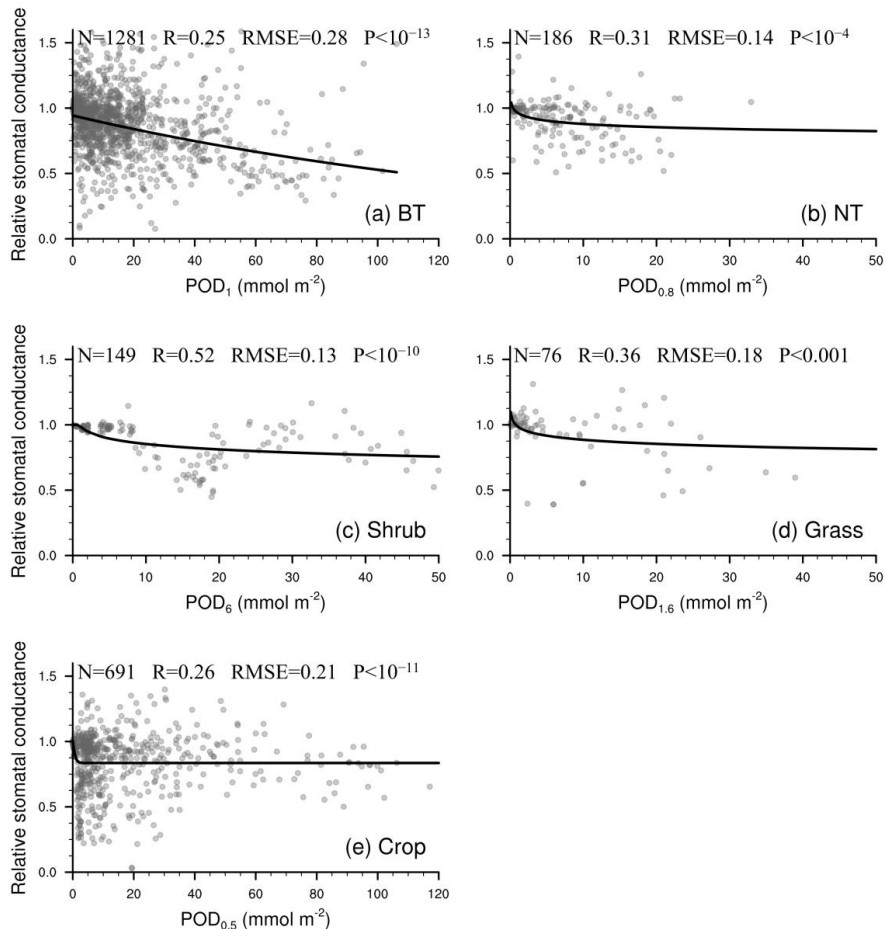

**Figure 4.** Same as Fig. 3, but for stomatal conductance.


**Table 3**. Overview of improved ability from the L15 scheme to the new scheme in reproducing the observed relationship between $POD_Y$ and either relative photosynthetic rate or relative stomatal conductance for various vegetation types, based on the database collected in our study. NS: non-significant, *: $P<0.05$, **: $P<0.01$, ***: $P<0.001$. When both schemes are significant, we also list the

relative changes in $R^2$ of the new scheme to L15.

| Veg. type | Photosynthetic rate | Stomatal conductance |
|---|---|---|
| BT | NS→*** | NS→*** |
| NT | NS→*** | NS→*** |
| Shrub | NS→*** | NS→*** |
| Grass | NS→** | NS→*** |
| Crop | Both ***, New: + 8.1% $R^2$ | NS→*** |





$POD_Y$ (mmol m$^{-2}$) in Eqs. (2) and (3) represents the cumulative $O_3$ uptake during the vegetation growing season. Its value in timestep $t$ is calculated as:

$$POD_{Y,t} = POD_{Y,t-1}(1-D_t) + U_{Y,t} \times 10^{-6},\qquad(4)$$

where $POD_{Y,t}$ and $POD_{Y,t-1}$ are the $POD_Y$ at timesteps $t$ and $t-1$; $D_t$ (0 to 1, unitless) is the decay

fraction at timestep $t$ given that leaves fall and emerge as well as $POD_Y$ in process-based models

represent the PFT average in a grid cell; $U_{Y,t}$ (nmol $O_3$ m$^{-2}$ timestep$^{-1}$) is the daytime $O_3$ uptake at

timestep $t$; $10^{-6}$ is the unit converter from nmol to mmol. The growing season is defined as leaf area

index (LAI, m$^2$ m$^{-2}$) > 0.3 for temperate deciduous shrubs and LAI > 0.5 for other deciduous PFTs, and

all year for evergreen PFTs. The LAI threshold of 0.5 is used by Lombardozzi et al. (2015). For the

temperate deciduous shrubs, a threshold of 0.5 is too high and close to the observed peak month LAI

according to CLM5 present-day surface data (generated from the MCD15A LAI product, Lawrence et

al., 2019), so we use a lower value of 0.3 as the threshold.

The decay fraction is set as:

$$D_t = \begin{cases} \dfrac{\Delta t}{l_{leaf} \times 3600*24*365} & \text{evergreen} \\[2ex] \max(0,\ 1-\dfrac{LAI_{t-1}}{LAI_t}) & \text{else} \end{cases},\qquad(5)$$

where $\Delta t$ is timestep length (sec); $l_{leaf}$ is leaf longevity (yr); $LAI_{t-1}$ and $LAI_t$ are leaf area index at

timesteps $t-1$ and $t$, respectively. $l_{leaf}$ is set to 1.7, 3.2, 1.3, and 6.5 years for tropical broadleaf

evergreen trees, temperate needleleaf evergreen trees, temperate broadleaf evergreen trees, and boreal

needleleaf evergreen trees, respectively, according to Zhang et al. (2016) which assessed the leaf

longevity based on 418 field measurements around the world. The leaf longevity value (1.3 years) of

temperate broadleaf evergreen trees is used for temperate broadleaf evergreen shrubs. For evergreen

PFTs, the function of $D_t$ is typically used to calculate the leaf turnover rate in DGVMs. For deciduous

PFTs, we consider the decay of cumulative $O_3$ uptake during the green-up period. We prefer the

function of LAI over leaf carbon pool for broader application because (i) land surface models and

ESMs often run with prescribed vegetation and inactive carbon cycle module (Dai et al., 2013, 2020;

Lawrence et al., 2019; Song et al., 2021), and (ii) many DGVMs update carbon pools at the end of a

year while updating LAI daily so they do not model the changes in leaf carbon during the growing

season, e.g., LPJ-DGVM, CLM-DGVM, IAP-DGVM, and CoLM-DGVM (Sitch et al., 2003; Levis et

al., 2004; Zeng et al., 2013; Ji et al., 2014). For models with carbon pools updated at a sub-hourly to





daily timestep, an alternative function of $D_t$ for deciduous PFTs is to use leaf carbon to replace LAI.

The $O_3$ uptake at timestep $t$ is calculated using:

$$U_{Y,t} = \begin{cases} \Delta t \times \max(F_{O3,t} - Y, 0) & \text{daytime} \\ 0 & \text{else} \end{cases},$$ (6)

where ozone flux threshold $Y$ (nmol $O_3$ m$^{-2}$ s$^{-1}$) is 3 for BT, 1 for NT, 5 for shrub, 2 for grass, and 0.5 for crop based on Sect. 2.2.2; the instantaneous $O_3$ flux to stomata at timestep $t$, $F_{O3,t}$ (nmol $O_3$ m$^{-2}$ s$^{-1}$), is estimated in analogy with Ohm's law by:

$$F_{O3,t} = \frac{[O_3]_t}{r_{b,t} + r_{am,t} + r_{s,t} k_{O3}},$$ (7)

where $[O_3]$ is the $O_3$ concentration at reference level (nmol m$^{-3}$); $r_{am}$ (s m$^{-1}$), $r_b$ (s m$^{-1}$), and $r_s$ (s m$^{-1}$) are aerodynamical resistance, boundary layer resistance, and leaf stomatal resistance, respectively. Eq. (7) is similar to S07 and L15 but with the updated value of $k_{O3}$.

After response factors are calculated based on Eqs. (2) and (3), the leaf net photosynthetic rate ($A_{n,t}$, μmol m$^{-2}$ s$^{-1}$) and stomatal conductance ($g_{s,t}$, μmol m$^{-2}$ s$^{-1}$) at timestep $t$ are modified for ozone stress as

$$A_{n\_O3,t} = A_{n,t} \times F_{O3\_A,t}$$ (8)

And

$$g_{s\_O3,t} = g_{s,t} \times F_{O3\_g,t}.$$ (9)

In process-based models, net photosynthetic rate $A_n$ is the photosynthetic rate minus dark respiration, where the photosynthetic rate is usually calculated using the Farquhar-Collatz model (Farquhar et al., 1980; Collatz et al., 1992). Stomatal conductance $g_s$ is generally estimated according to the Medlyn (Medlyn et al., 2011) or Ball-Berry (Ball et al., 1987; Collatz et al., 1991) models. $CO_2$ partial pressure at the leaf surface and in the leaf, vapor pressure at the leaf surface, stomatal resistance (the reciprocal of stomatal conductance), and net photosynthetic rate are solved iteratively. The impact of $O_3$ plant damage is not considered during the iterations.

## 4 Application

### 4.1 $O_3$ effect on global leaf photosynthetic rate and stomatal conductance

We integrate the new scheme into the CESM2.2's land component CLM5, to quantify the impact of

ozone exposure on global leaf photosynthetic rate and stomatal conductance for 2005–2014. The

growing-season average of daytime $O_3$ concentration is high mainly in the mid-latitudes (20–50° N) of

the Northern Hemisphere (NH) (Fig. 5a). The areas with the highest $O_3$ concentrations are in the

western United States, western and central Asia, and northern Africa, largely coinciding with the NH

arid and semi-arid regions. $O_3$ concentrations over boreal grasslands and shrublands as well as tropical

savannas are higher than those in the tropical rainforests in South America (i.e., Amazon rainforest),

Africa (i.e., Congo rainforest), and New Guinea, but lower than those in NH forests and croplands. The

peak-month $O_3$ concentrations during the growing season are much higher than the growing season

average, overall exceeding 40 ppb across most vegetated regions (Fig. 5b).

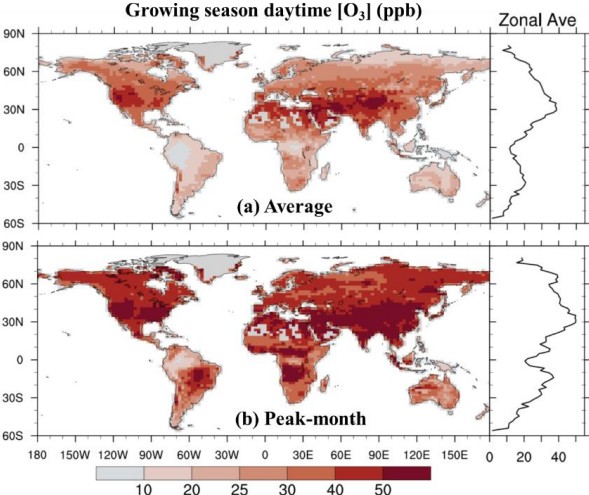

**Figure 5**. 2005−2014 average of (a) the growing season average of daytime surface $O_3$ concentration and

(b) the highest monthly concentration during growing season.

Annual cumulative $O_3$ uptake for sunlit leaves is high over the temperate forests and croplands in

East Asia, Southeast Asia, South Asia, United States, and Europe, as well as the boreal evergreen forest

zone around 55 °N (Fig. 6a). Most of these regions are those with moderate to high $O_3$ concentrations

(Fig. 5a) or long growing season. Low-value regions are characterized by either low $O_3$ concentrations,

such as in the heart of the Amazon and Congo rainforests, or low stomatal conductance, such as in NH

temperate arid regions due to dry conditions and in boreal grasslands and shrublands due to the cold

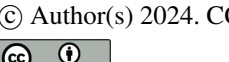



climate. The spatial pattern for shaded leaves is similar but with much lower values due to lower

stomatal conductance (Fig. 6b).

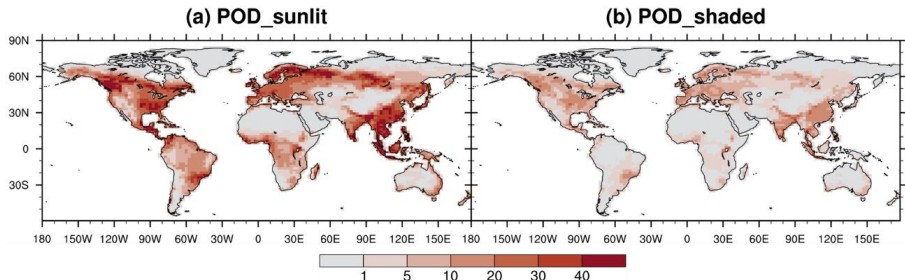

**Figure 6.** Annual average $POD_Y$ (mmol m$^{-2}$) for (a) sunlit and (b) shaded leaves in O$_3$_New

simulations.

As shown in Fig. 7, O$_3$ significantly reduces annual leaf photosynthetic rate and stomatal

conductance over most vegetated areas, with a global average reduction of 8.5% for the former and

7.4% for the latter, both significant at the 0.05 level according to the student's t-test. The spatial pattern

of O$_3$-induced significant reduction in leaf photosynthetic rate (Fig. 7a) is similar to that of sunlit-leaf

cumulative O$_3$ uptake (Fig. 6a). O$_3$-induced reduction in stomatal conductance is typically weaker, with

the largest reductions located in East Asia, Southeast Asia, and South Asia (Fig. 7b).

Compared to the new scheme, the L15 scheme generally simulates a stronger reduction in both

photosynthetic rate and stomatal conductance (Figs. 7c–d), particularly in the tropical savannas across

South America, Africa, and Australia and in the grasslands and shrublands over boreal Asia for

photosynthesis (Fig. 7e) and the tropical savannas across Africa, South America, and Australia for

stomatal conductance (Fig. 7f). The estimated global reduction is 20.4% for leaf photosynthesis and

13.4% for stomatal conductance. Both reductions are statistically significant at the 0.05 level, and are

2.4 and 1.8 times greater, respectively, than those estimated with the new scheme.



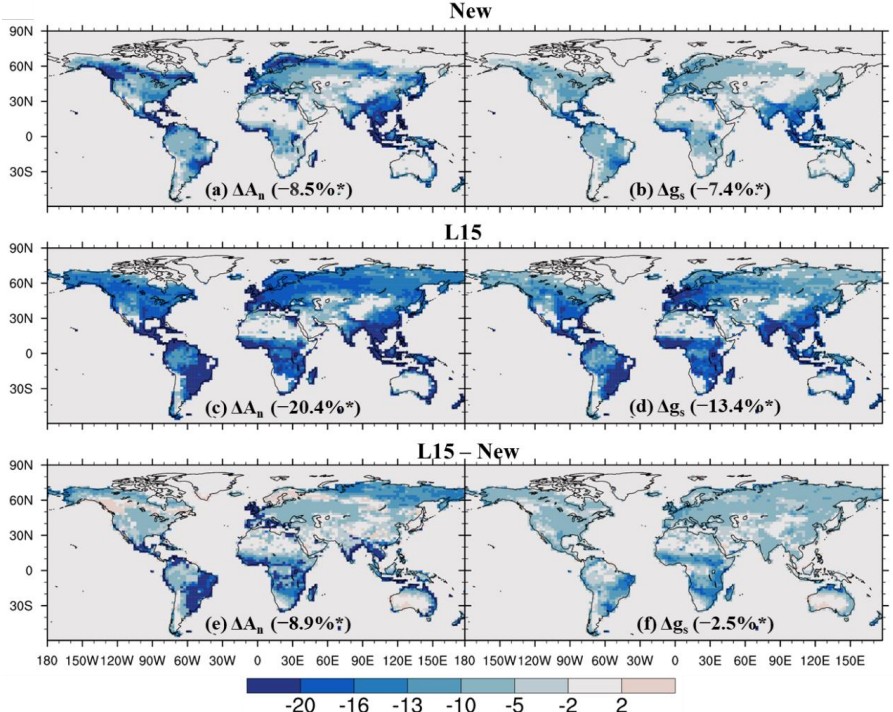

**Figure 7.** Relative impact (%) of $O_3$ on net leaf photosynthetic rate ($A_n$) and stomatal conductance ($g_s$) quantified using (a−b) the new and (c−d) L15 schemes, as well as (e−f) the difference between them. In (a−d), the relative impacts are calculated using $O_3$_New or $O_3$_L15 compared to $O_3$_OFF; only areas where the $O_3$ impact is statistically significant at the 0.05 level are shown; numbers in parentheses are the global average influence. * Indicates that the (a−d) global influence and (e−f) their difference is significant at the 0.05 level.

The influence of $O_3$ differs widely among PFTs, ranging from 0–17.1% for photosynthetic rate and 0–15.7% for stomatal conductance. Crops and trees are the most affected, followed by grasses, and shrubs are the least affected (Fig. 8). Grasses and shrubs are less affected mainly due to their lower cumulative $O_3$ uptakes. Among trees, evergreen PFTs are more responsive to $O_3$ than their deciduous counterparts within needleleaf or broadleaf types, attributable to their longer growing season and thus longer $O_3$ exposure and higher cumulative $O_3$ uptake. The photosynthetic rate of temperate broadleaf trees and boreal broadleaf deciduous trees is more affected than that of temperate needleleaf trees and boreal needleleaf deciduous trees (Fig. 8a) due to the higher sensitivity of broadleaf versus needleleaf





photosynthesis (Eq. 2 and Fig. 3). Broadleaf trees and grasses exhibit a greater photosynthetic response

than stomatal response (Fig. 8), highlighting the importance of nonstomatal $O_3$ response mechanisms

for photosynthesis, e.g., $O_3$ decreases photosynthesis by reducing the mesophyll conductance in

observations (Xu et al., 2023).

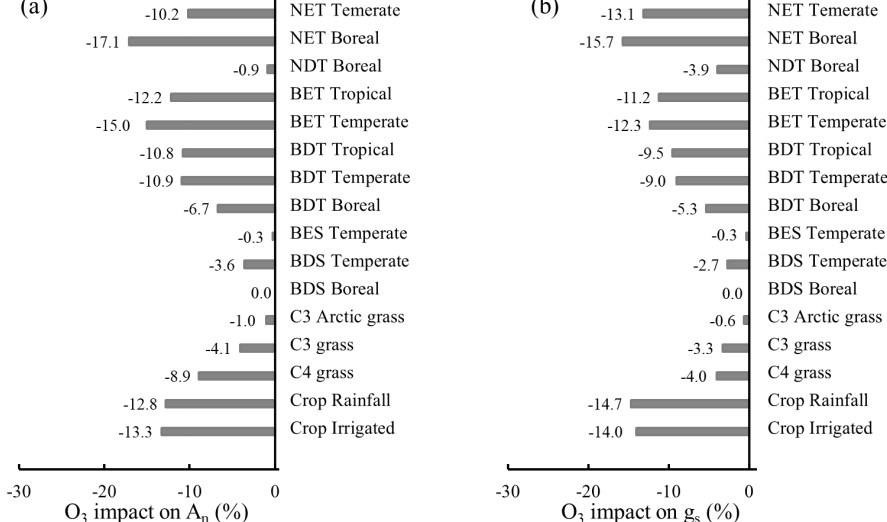

**Figure 8.** Global PFT-level relative impact (%) of 2005−2014 $O_3$ exposure for (a) $A_n$ and (b) $g_s$,

quantified by $[(O_3\text{-New} – O_3\_OFF) / O_3\_OFF] \times 100\%$. Abbreviations: T: tree; S: shrub; N: needleleaf;

B: broadleaf; E: evergreen; D: deciduous. CLM5 PFTs are used and their global distribution is shown

in Fig. S1.

On seasonal cycle, the impact of $O_3$ on the seasonal phase of both leaf photosynthetic rate and

stomatal conductance is small, shifting the peak month by less than one month in most regions (Figs.

9a–b). However, $O_3$ exerts a strong influence on the magnitude of seasonal cycle (Figs. 9c–d). It

decreases the seasonal amplitude of photosynthetic rate in mid- and low-latitude vegetated areas except

in evergreen forests (Fig. 9c). For stomatal conductance, the reduction is even greater and more

widespread (Fig. 9d). Areas with up to a 50% reduction in stomatal conductance include Eastern North

America, Europe, East Asia, South Asia, and tropical savannas in North Africa. This dampening of

seasonal variation mainly due to the partial overlap between peak periods of photosynthesis and




stomatal conductance and those of cumulative $O_3$ uptake because the latter is influenced by stomatal

conductance.

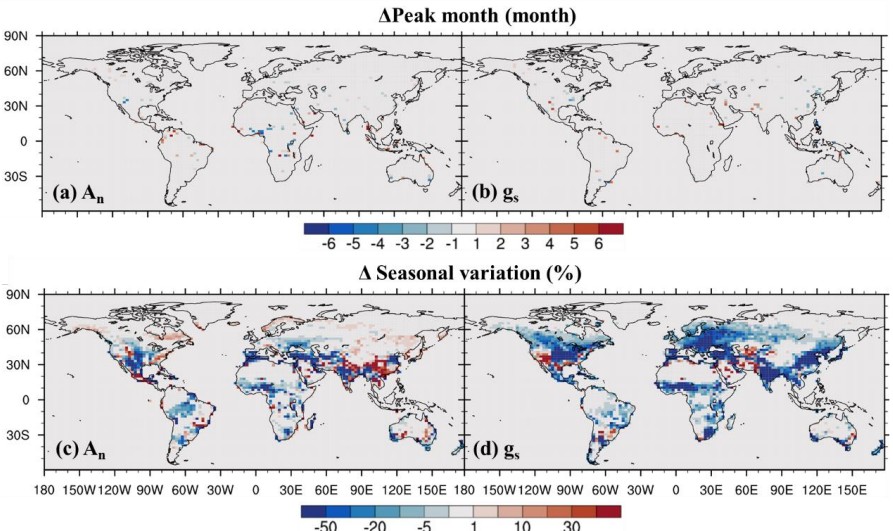

**Figure 9.** $O_3$ impact on (a−b) peak month and (c−d) seasonal amplitude quantified by the Coefficient of

Variation.

## 4.2 Effects on global GPP simulations

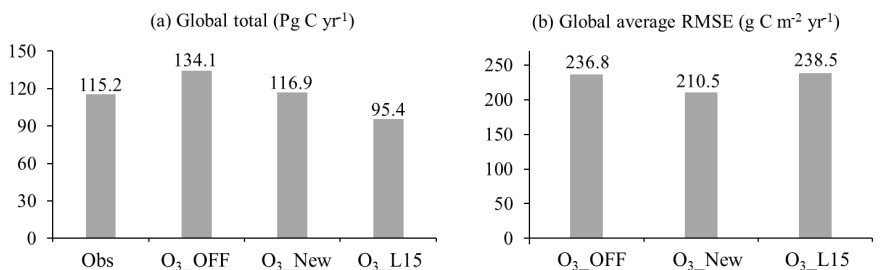

**Figure 10.** 2005−2014 averaged (a) global total Gross Primary Production (GPP) of FLUXCOM (Obs)

and simulations, and (b) global land average of Root Mean Square Error (RMSE) of GPP between

FLUXCOM and simulations.

$O_3$ plant damage, as quantified using the new scheme, decreases the global GPP from 134.1 to 116.9 Pg

C yr$^{-1}$ (a 12.8% reduction) for the period 2005 to 2014 (Fig. 10a). The global total GPP simulated with




the new scheme aligns closely with the FLUXCOM benchmark (115.2 Pg C yr$^{-1}$). The global average

RMSE between simulations and FLUXCOM is reduced by 11.1% (Fig. 10b) compared to the

simulations without O$_3$ plant damage, justifying the significance of incorporating O$_3$ plant damage into

large-scale process-based models.

In comparison, the L15 scheme estimates a very strong O$_3$-induced decrease in GPP, up to 28.9%,

yielding a global GPP estimates (95.4 Pg C yr$^{-1}$) much lower than the FLUXCOM (Fig. 10a).

Furthermore, the RMSE is 238.5 g C m$^{-2}$ yr$^{-1}$, which is close to the value of the simulation without O$_3$

plant damage. The RMSE of the new scheme is 11.7% lower than that of the L15 scheme,

demonstrating the superiority of the new scheme over the L15 scheme (Fig. 10b).

Spatially, incorporating the new scheme improves simulations by reducing the overestimation of

GPP over the boreal forest zone around 55 °N, tropical savannas, and American croplands (Figs. S2a

and S3a–b). It also lessens the underestimation of GPP over Europe, East and West America, South

America, African rainforests, East Asia, Southeast Asia, and South Asia in L15 simulations (Figs. S2b

and S3b–c).

**5 Conclusions and discussion**

**5.1 The new parameterization scheme**

**5.1.1 Summary**

This study proposes a new parameterization scheme designed to integrate the response of leaf

photosynthetic rate and stomatal conductance to O$_3$ exposure into process-based models (e.g., land

surface models, DGVMs, GGCMs, or ESMs), enabling regional and global simulations of O$_3$ plant

damage and its subsequent influence. The scheme is built using the most comprehensive compilation of

observations gathered from peer-reviewed literature. Functions of flux-based ozone index POD$_Y$ are

found out to accurately reproduce the statistically significant linear and nonlinear relationships between

POD$_Y$ and either relative leaf photosynthetic rate or stomatal conductance in observations for

needleleaf trees, broadleaf trees, shrubs, grasses, and crops.

**5.1.2 Advantages**

The new parameterization scheme exhibits obvious advantages over previous parameterization



schemes. First, it is built on 4210 paired data points from $O_3$ fumigation experiments, over six times of those employed in earlier schemes. We extend data collection from peer-reviewed literature to

December 2022, compared to June 2011 in L15 and before 2004 in Felzer et al. (2004) and S07. The comprehensive dataset enhances the representation of the new scheme and supports the response functions established for shrubs (and grasses) which previously used the observed responses for trees (and crops) in L15 (and S07) due to a lack of observations. Also, the data we compiled are observed photosynthetic and stomatal responses rather than biomass or yield responses which were the

foundation of S07. This way we need not estimate the parameters of photosynthetic and stomatal responses through the inverse method used in S07 to fit the observed yield or biomass response, thereby the response functions and parameters in the new scheme are model- and bias-independent, which enhances the accuracy and applicability.

Second, it accurately reproduces statistically significant linear or nonlinear photosynthetic and

stomatal response to O3 in observations for all the vegetation types, eliminating the need to apply the response function of one vegetation type to another or to use constants. The L15 scheme, which assumes a linear response, was only able to reproduce the observed relationship with $POD_Y$ for only the crop photosynthetic rate and temperate evergreen tree stomatal conductance. When evaluated with our expanded observations, applying the response function of temperate evergreen tree stomatal

conductance to needleleaf trees by L15 is found to be unsupported (Table 3).

The nonlinear functions built for most vegetation types in the new scheme depict a decreasing plant sensitivity with increasing $POD_Y$, different from the constant sensitivity implied by linear functions. Our observation dataset aggregates data from diverse plant species into broader vegetation types and demonstrates the decreased sensitivity. This decrease in sensitivity reflects the plant

adaptability or a transition from sensitivity to tolerance among plant species naturally (e.g., competition) or anthropogenic (e.g., genetic variation, breeding) in the real world (Fuhrer, 2003; Frei et al., 2014; Agathokleous et al., 2020). Current global process-based models do not simulate such adaptability and are limited to representing PFTs without differentiation among plant species (Bonan, 2019). The nonlinear response functions we have developed will enable these models to capture the

variability in plant ozone tolerance and the shift among plant species for both intra- and inter-PFT within a vegetation type, despite not directly modeling species-level responses.

In addition, the new scheme sets the photosynthetic and stomatal responses as a function of $POD_Y$.





In contrast to the product of stomatal conductance and AOT40 used in Felzer et al. (2004), $POD_Y$ has a
clear physical interpretation, considering not only high $O_3$ concentrations but also chronic ozone

exposure at moderate or low $O_3$ levels. Compared to S07, this scheme provides an optimal
representation of $O_3$ plant damage rather than upper and lower response thresholds, aligning with other
processes represented in process-based models. Moreover, like L15, our scheme considers the
decoupling of stomatal conductance and photosynthetic rate under ozone exposure, an observational
fact not accounted for in Felzer et al. (2004) and S07.

**5.1.3 Implementation**

The new scheme has important potential for both academic research and practical implementation.
First, it is important for the development of large-scale process-based models. Although S07 and L15
have been integrated into JULES and CLM (the land components of UKESM and CESM,
respectively), they are not active in default runs (Lawrence et al., 2019) partly due to limited

representation of observations. Our scheme offers considerable improvements, detailed in Sect. 5.1.2,
enabling process-based models to reasonably simulate the observed $O_3$ plant damage. Our results also
show that, when using CESM2.2's CLM5, the new scheme reduces global GPP simulation bias by
11.1% compared to simulations without $O_3$ plant damage, and by 11.7% compared to the old scheme
(i.e., L15), underscoring the necessity of incorporating $O_3$ plant damage into large-scale process-based

models and the utility of our new scheme.

Second, it can improve our understanding and projection accuracy of the role of $O_3$ plant damage
in the Earth system on regional and global scales. Rising $O_3$ is currently a critical environmental issue
in the world. Even though many studies quantified its impacts using various models, they mainly
focused on GPP, NPP or a specific region and their results are highly uncertain. We have already

developed a new parameterization scheme in this study. Moving forward, we will comprehensively
quantify the influence of $O_3$ plant damage on ecosystems and climate using ESMs equipped with the
new scheme, as we did for wildfires, another important form of terrestrial ecosystem disturbance (Li et
al., 2014, 2017, 2019, 2021; Jiang et al., 2016; Li and Lawrence 2017; Lasslop et al., 2020).

In addition, the new scheme aids in establishing an effective model platform to calculate the

impact of proposed industrial developments, emissions standards, and land use changes on ecosystems,
climate, and socioeconomics, guiding the formulation of effective policies for air quality control,



climate mitigation, and biodiversity conservation.

### 5.1.4 Future development

There are four potential directions for further development. First, besides the average of a sample (e.g.,

multiple measurements, measurements on different leaves or different individuals), the observation

dataset we compiled contains sample size, standard deviation (SD), and standard error (SE) for most

data points. Incorporating the additional information allows us to assign greater weight to data points

that are more reliable, such as those with larger sample sizes and/or smaller SD or SE, thereby

enhancing the representativeness of the response functions.

Second, this study only tests the commonly used linearizable nonlinear functions. Other two-

parameter nonlinear functions may better capture the photosynthetic and stomatal responses.

Third, introducing other explanatory variables may reduce the number of parameters that require

estimation. Karlsson et al. (2007) and Bussotti (2008) found that plant sensitivity to $O_3$ was linked to

leaf morphological traits like leaf area, thickness, and leaf mass per area (LMA). Feng et al. (2018)

further suggested using LMA to unify the response of woody species to $O_3$ and proposed a function of

trait-based ozone plant sensitivity. Ma et al. (2023) combined the function with S07 and tested it in a

DGVM and verified that using one unified sensitivity parameter for all PFTs and observed global LMA

map could yield results similar to S07 which uses multiple vegetation-type-dependent parameters. Yet,

it is important to consider the inherent simulation uncertainty and bias of the new explanatory variables

and their influence and if the approach works for all vegetation types and species.

Conversely, some researchers strive to further subdivide vegetation or crop types for more

accurate fitting (Singh et al., 2023; Guarin et al., 2023). However, the current experimental data for $C_4$

crops and tropical plants are limited and may not adequately support the detail categorization from the

perspective of big data for big ecology. Especially as the variety of vegetation and crop types continues

to grow in process-based models, the demand for observations will likely grow.

Our database offers the most comprehensive compilation of observations to date, supporting the

above development directions and enabling their evaluation, selection, and integration.

### 5.2 Global impact assessment using the new scheme

As an application example, we integrate the new scheme into CESM2.2's land component CLM5 to





assess the global physiological impact of $O_3$ exposure from 2005 to 2014. This is done by quantifying

the difference between simulations with and without $O_3$ plant damage. Our results indicate that present-

day $O_3$ exposure leads to an 8.5% reduction in global leaf photosynthetic rate and 7.4% reduction in

stomatal conductance, and spatially with largest reduction in eastern and southern Asia, Europe, eastern

United States, and the boreal evergreen forests zone for the former and in the eastern and southern Asia

for the latter. These results, at a global scale, supports the experiment results that chronic $O_3$ exposure

decouples the photosynthetic rate and stomatal conductance (Tjoekler et al., 1995; Wittig et al., 2007;

Lombardozzi et al., 2012; Kinose et al., 2020). We also estimate a 11.3% and 10.5% reduction in

photosynthetic rate and stomatal conductance for trees, similar to 11% and 13% estimated by Wittig et

al. (2007) based on a meta-analysis of a smaller observational dataset. When examining the effects at the

PFT-level, we found that crops are most affected, followed by trees, with grasses intermediate and shrubs

least affected. Ma et al. (2023) also reported crop was most affected under present-day $O_3$ concentration

quantified using YiBs with the S07-LMA scheme. Interestingly, as far as we know, this study is the first

to discover that $O_3$ exposure generally leads to a decrease in seasonal amplitude over most vegetated

areas, especially for stomatal conductance, while only causing limited changes in their seasonal pattern.

In addition, using the new scheme, we estimate a global GPP reduction of 12.8% due to $O_3$, which

is less than half of the 28.9% reduction estimated using L15 in the CLM5. The discrepancy arises L15

using lower flux thresholds $Y$ for broadleaf trees, shrubs, and grasses, as well as functions representing

an overall higher sensitivity to $O_3$ for crops, needleleaf trees, and grasses, considering the nighttime $O_3$

uptake, and limiting the impact of leaf fall and emergence to the ozone uptake at a single time-step (i.e.,

$U_{Y,t}$) (See Appendix). Our estimate is higher than the quantification result of S07 (2−5%, Yue and

Unger, 2015) and S07-LMA (4.8%, Ma et al., 2017) in YiBs, but lower than L15 in CLM4.5 (10.8%)

(Lombardozzi et al., 2015) and in CLM5 (28.7%), the influence of $O_3$ estimates by the new scheme

likely lies between S07 and L15 if using the same model platform. The big disparity in the estimated

influence of L15 between CLM5 and CLM4.5 suggests the potential benefit of employing multiple

process-based models to quantify the uncertainty of $O_3$ influence due to the different stomatal

conductance across models which will affect the estimated $POD_Y$. For example, the inclusion of plant

hydraulic stress in CLM5 increases stomatal conductance, leading to higher $POD_Y$ and thus higher $O_3$

influence.



**5.3 Suggestions to the observational community**

Currently, an increasing number of $O_3$ fumigation experiments are exploring the relationship between $POD_Y$ and the crop yield or biomass of trees and grasses, which is beneficial for IAMs (CLRTAP, 2017). Nevertheless, modeling the dynamic responses of carbon, water, energy, and even climate is crucial for large-scale process-based models and for accurate projections of global change. Therefore, $O_3$ fumigation experiments that quantify the sensitivity of photosynthetic rates and stomatal

conductance are still necessary, particularly for $C_4$ crops and tropical plants, which remain underrepresented in observations. Furthermore, this study objectively establishes the optimal flux threshold of $Y$ based on extensive observations, rather than arbitrary assignment as in L15 or those based on a small number of observations as in CLRTAP (2017). The flux threshold of $Y$ can serve as a reference for future observational analyses of leaf photosynthetic and stomatal responses. In addition,

parameterization schemes (including ours) often assume that the response relationship of a specific plant is the same for shaded and sunlit leaves. The assumptions must be validated or adjusted to a more reasonable ratio based on additional observations.

**Appendix A**

In the scheme proposed by Lombardozzi et al. (2015, L15) and used in CLM5, the response factor to $O_3$ for photosynthetic rate is:

$$F_{O3\_A} = \begin{cases} 0.8752 & \text{Broadleaf tree \& shrub} \\ 0.8390 & \text{Needleleaf tree \& shrub} \\ \text{use Crop's} & \text{Grass} \\ -0.0009POD_{0.8}+0.8021 & \text{Crop} \end{cases} \quad (A1)$$

and that for stomatal conductance is:

$$F_{O3\_g} = \begin{cases} 0.9125 & \text{Broadleaf tree \& shrub} \\ 0.0048POD_{0.8}+0.7823 & \text{Needleleaf tree \& shrub} \\ \text{use Crop's} & \text{Grass} \\ 0.7511 & \text{Crop} \end{cases} \quad (A2)$$

where $POD_{0.8}$ is phytotoxic $O_3$ dose over a threshold of 0.8 nmol $O_3$ m$^{-2}$ s$^{-1}$ during the growing season (defined as leaf area index LAI > 0.5 m$^2$ m$^{-2}$). When used in CLM5, the response factors in Eqs. (A1−2) are required to range from 0 to 1 to avoid unwanted outcomes in any scenario. Shrubs used the response functions of trees due to the unavailability of observations, while, for grasses, broadleaf trees,

and needleleaf trees, L15 employs the functions of crops, temperate deciduous trees, and temperate

evergreen trees, respectively, because significant linear regression functions were not found.

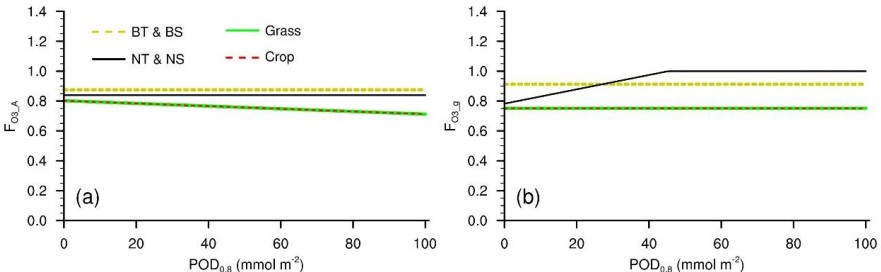

**Figure A1**. Response factors of (a) photosynthetic rate and (b) stomatal conductance to $O_3$ in L15

when used in CLM5. BT: broadleaf tree, BS: broadleaf shrub; NT: needleleaf tree; NS: needleleaf

shrub.

The value of $POD_{0.8}$ at time step $t$ is:

$$POD_{0.8,\,t} = POD_{0.8,\,t-1}(1-D_t)+U_{0.8,\,t}\times 10^{-6}. \tag{A3}$$

In Eq. (A3), the decay factor (0 to 1, unitless) is:

$$D_t = \begin{cases} \dfrac{\Delta t}{l_{\text{leaf}}\times 3600*24*365} & \text{evergreen} \\ 0 & \text{else} \end{cases}, \tag{A4}$$

where $\Delta t$ is timestep length and $l_{leaf}$ (year) is the leaf longevity.

The $O_3$ uptake at timestep $t$ is calculated using:

$$U_{0.8,t} = \Delta t\times \max(F_{O3,t}-0.8,0)(1-H). \tag{A5}$$

Here, the instantaneous $O_3$ flux to stomata at timestep $t$, $F_{O3,t}$ (nmol $O_3$ m$^{-2}$ s$^{-1}$), is calculated as Eq. (4),

and the heal factor $H$ (0 to 1, unitless) is set as:

$$H = \max(0,\ 1-\frac{LAI_{t-1}}{LAI_t}), \tag{A6}$$

where $LAI_{t-1}$ and $LAI_t$ are leaf area index at timesteps $t-1$ and $t$, respectively.

*Code and data availability.* The code and data will be available on Zenodo after manuscript acceptance.


*Autor Contributions.* FL conceived the research ideas, constructed the new parameterization scheme,



developed the model code, and performed the simulations and data analysis. ZMZ and FL collected

data from peer-reviewed literature. Data pre-processing was carried out by ZYZ, YZ, and FL. FL wrote

the manuscript draft. SL, SS, FH, ZF, and PBR reviewed and edited the manuscript.


*Competing interests.* The authors declare that they have no conflict of interest.

*Acknowledgements.* This study is co-supported by the National Natural Science Foundation of China

(41875137), Guangdong Major Project of Basic and Applied Basic Research (2021B0301030007),

National Key Research and Development Program of China (2022YFE010650), and the National Key

Scientific and Technological Infrastructure project "Earth System Science Numerical Simulator Facility"

(EarthLab). SS and FH are supported by UKRI National Environmental Research Council

NE/R001812/1. PBR is supported by the National Science Foundation: Biological Integration Institutes

NSF-DBI-2021898. For open access, the authors have applied a "Creative Commons Attribution (CC

BY) license to any Author Accepted Manuscript version arising". We are grateful to Danica Lombardozzi,

Zhongda Lin, Xu Yue, and Dezhen Yin for their discussions, Huanhuan Sun and Yue Hu for their

assistance with data collection and pre-processing, and Editor Hisashi Sato for helpful suggestions and

the time dedicated to handling the paper review process. We would also acknowledge the National Center

for Atmospheric Research (NCAR), principally funded by the US National Science Foundation (NSF)

under cooperative agreement no. 1852977, for providing Earth system model CESM2.2 code and input

data.

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
