# Peer review of "Quantifying the role of ozone-caused damage to vegetation in the Earth system: A new parameterization scheme for photosynthetic and stomatal responses"

_Geoscientific Model Development, 2024_

## Referee Comment (RC2)

**Peer-Review of**
**"Quantifying the role of ozone-caused damage to vegetation in the Earth system: A new parameterization scheme for photosynthetic and stomatal responses"**

**Title:** Quantifying the role of ozone-caused damage to vegetation in the Earth system: A new parameterization scheme for photosynthetic and stomatal responses
**Authors:** Fang Li, Zhimin ZHou, Samuel Levis, Stephen Sitch, Felicity Hayes, Zhaozhong Feng, Peter B. Reich, Zhiyi Zhao, Yanqing Zhou
Reviewed by Ye Liu

**Summary**

This manuscript presents a comprehensive study on the development of a new parameterization scheme to quantify the damage caused by surface ozone (O3) on vegetation. The proposed scheme integrates extensive observations from O3 fumigation experiments to improve the accuracy of simulating photosynthetic and stomatal responses in large-scale models. The authors compiled 4210 paired data points from peer-reviewed literature, significantly expanding the dataset used in previous studies. The new scheme demonstrates improved performance in the Community Earth System Model (CESM2.2) compared to earlier schemes, showing a reduction in global photosynthetic rate and stomatal conductance estimates, and enhancing the simulation of global Gross Primary Production (GPP).

This manuscript advances this topic with the following developments: (1) Development of a new parameterization scheme based on an extensive dataset, which is over six times larger than those used in previous studies. (2) Improved accuracy in simulating the response of various vegetation types to O3 exposure, including needleleaf trees, broadleaf trees, shrubs, grasses, and crops. (3) Incorporation of nonlinear response functions that reflect the decreasing sensitivity of plants to increasing O3 doses, capturing the variability in plant ozone tolerance. (4) Integration of the new scheme into CESM2.2, resulting in more accurate simulations of global leaf photosynthetic rate, stomatal conductance, and GPP.

**Major Remarks**

I understand there is generally a lack of observational data to validate many levels of processes related to plant photosynthesis. The manuscript could benefit from a more detailed discussion of the potential uncertainties and limitations associated with the observational data and model implementation.

1. Figure 5 shows the simulated O3 concentration. Is there any way to quantify the uncertainties and how this uncertainty would affect the impact of POD?

2. Figures 3 and 4 set the foundation of the new scheme, as mentioned by Reviewer #1, the spread of the data point suggesting POD may not be the dominant factor driving the photosynthetic response. For instance, the fitted line for crop indicates a large reduction in photosynthetic rate with POD increase from 0 to 10 mmol m-2. Considering the large spread of the data points within this POD range, the concern would be this fitting line could be associated with large uncertainties. This uncertainty can propagate in the model and affect the O3 impact GPP, An, and gs analysis, such as the large reduction of An and gs of the crop in response to O3 in Figure 8.

**Specific Remarks**

1. BT and NT are defined in Line 336, but used before this line.

2. Line 449-450: The manuscript states that the global reduction in leaf photosynthetic rate and stomatal conductance is 8.5% and 7.4%, respectively. It would be useful to provide a comparison with previous estimates from earlier schemes, either at global or regional scales, if any.

---

## Author Comment (AC4)

1. Figures 3 and 4: I acknowledge that these data are sourced from diverse literature and encompass a variety of environmental conditions and locations. Consequently, expecting a clear linear relationship might be unrealistic. Nonetheless, given the big unexplained variation, there may be potential to improve the fitting. I wonder if incorporating environmental factors or applying region-specific fitting functions could better capture the photosynthetic response. I recommend that the authors discuss potential strategies to account for the remaining variation in the data.
   **Reply:** Introducing other explanatory variables or doing PFT/biome/regional fitting rather than the current broader vegetation-category fitting may improve the fitting skill (i.e., explain the remaining variation in the data).

   In the revised version, we have acknowledged the limitations of the new scheme by adding "Even though the new scheme has advantages over earlier schemes, as listed in the previous section, there are still noticeable variations in observations that have not been explained (Figs. 3 and 4).".

   We have also added your suggestions in potential development directions 3 and 4 in Sec. 5.1.4, which discuss the new scheme's future development as follows.

   In direction 3 (introducing other explanatory variables), we have added: "Furthermore, earlier studies found that environmental factors (e.g., $CO_2$ concentration, nitrogen availability, drought, and temperature) can influence the $O_3$ photosynthetic response through changing POD (e.g., Wittig et al. 2007; Hansen et al. 2019; Xu et al., 2020). These factors may also affect the relationship between POD and $O_3$ photosynthetic response, although there have been no analyses to verify this and identify the underlying mechanisms. Based on our dataset and by collecting data on environmental factors in corresponding experiments, we may be able to investigate this in the future. If the influence exists, introducing environmental factors will improve the fitting.".

   In direction 4, we have revised the first sentence to: "In addition, conducting PFT, biome, or regional fitting rather than the current broader vegetation type fitting may reduce the unexplained variation in observations.".

2. Line 63-67: The sentence is too long. Consider separating it into two sentences.
**Reply:** Separated.

3. Line 87: "physical" I would say "biophysical".
**Reply:** It has been revised to "biophysical" as you suggested.

4. Line 115-116: Better put the version number of CLM and CESM and mention whether the current implementation is on the same or different versions of CLM and CESM. So, the reader will know whether or not L15 is comparable with this ozone stress scheme.
**Reply:** We have included the version numbers (CLM5 and CESM2.2).

It's on the same version of CLM and CESM. In Section 2.3.2, we mentioned that the comparison between L15 and the new scheme, with results analyzed in Section 4 (Application), is based on the same model platform, input data, and protocol; only the O3 stress schemes are different.

5. Line 166-167: "only data categorized as high and medium confidence defined by Lombardozzi et al., (2013)" Need generally mention the confidence level is defined based on what standard in Lombardozzi et al., (2013).

**Reply:** We have added the definition of confidence level in Lombardozzi et al. (2013) as "In Lombardozzi et al. (2013), data were assigned high confidence if POD was presented, medium confidence if the publication contained multiple stomatal conductance measurements throughout the course of the experiment and other enough information to calculate POD, and low confidence otherwise".

6. Line 169-171: "if the data are previously or more completely reported in another article" do you mean the data is repeatedly reported?

**Reply:** Yes, it sometimes happens that the data is reported in multiple articles.

7. Line 290-291:"2000Clm50Sp" and "2000Clm45Sp" Better use a simple description rather than the CESM configuration abbreviation, which will be more friendly for those who don't use CESM.

**Reply:** Thanks for the suggestion. We have added "(present-day offline simulations of the land model CLM5.0 with prescribed vegetation)" after "I2000Clm50Sp," and "(present-day offline simulations of the land model CLM4.5 with prescribed vegetation)" after "I2000Clm45Sp," to make it more user-friendly for readers with varying familiarity with CESM.

8. Line 306: 1.9º should be 1.875º.

**Reply**: We have confirmed the latitude and "1.9°" has been revised to "1.895°".

9. Line 311: Missed one atmospheric forcing "Downward longwave radiation"

**Reply**: Thanks for pointing this out. We have added "incident longwave radiation".

10. Line 313-314: "have no interannual variability" mislead. MODIS data, of course, have interannual variability. Maybe you just want to say "you use a prescribed climatology of vegetation distribution and structure, which is based on present-day MODIS satellite observation"

**Reply**: We have changed the sentence to "The input data of the prescribed present-day vegetation distribution and structure (LAI and canopy height) have no interannual variability, which is derived from MODIS satellite observations.".

11. Line 319: "28.9655/47.9982" I'm not sure if we really need such high precision.

**Reply**: The unit conversion equation is provided by ECMWF, from which we obtained the $O_3$ concentration reanalysis data (EAC4) used as the input for CLM5.

Although the high precision may not be necessary, it does not cause any adverse effects.

**Reply**: According to your suggestion, we have revised it to "each vegetation type has its own function based on observations.".

**Reply**: Our results on $O_3$ affecting trees are similar to the Meta-analysis of Wittig et al. (2007), and that crops are most sensitive to $O_3$ is consistent with earlier observational analyses (Reich, 1987; Wang et al., 2024) and modeling works (Ma et al., 2023). The comparisons are discussed in Sec. 5.2, along with other comparisons with literature.

**Reply**: "Decouple" means that the $O_3$ influence on global photosynthetic rate and stomatal conductance differs. This is an extension of the conclusion of our scheme application results we presented before the sentence: "Our results indicate that present-day $O_3$ exposure leads to an 8.5% reduction in global leaf photosynthetic rate and a 7.4% reduction in stomatal conductance, with the largest reductions in eastern and southern Asia, Europe, the eastern United States, and the boreal evergreen forest zone for the former, and in eastern and southern Asia for the latter.". That is, the Sitch et al. (2007) scheme, which assumed the photosynthetic response equals stomatal conductance response, is not correct, partly because it misses the $O_3$ non-stomatal limitation to photosynthesis found in earlier mechanism analyses based on site-scale observations (described in our Introduction section).

To clarify this, we have revised the sentence to "Our results that $O_3$ influence on photosynthetic rate and stomatal conductance differs at a global scale support the findings of observational analyses that chronic $O_3$ exposure decouples the photosynthetic rate and stomatal conductance partly due to $O_3$ non-stomatal limitation to photosynthesis (Tjoekler et al., 1995; Wittig et al., 2007; Lombardozzi et al., 2012; Kinose et al., 2020).". In addition, Fig. 7 already clearly shows the decoupling, so we haven't added the scatter plot.

---

## Author Comment (AC5)

Major Remarks
The manuscript could benefit from a more detailed discussion of the potential uncertainties and limitations associated with the observational data and model implementation.
1.  Figure 5 shows the simulated $O_3$ concentration. Is there any way to quantify the uncertainties and how this uncertainty would affect the impact of POD?
**Reply:** I apologize for any confusion caused. In this study, the $O_3$ concentration data were not simulated by CLM5. Instead, we used 3-hourly 0.75º $O_3$ concentration data from the ECMWF Atmospheric Composition Reanalysis 4 (EAC4) as input for CLM5, as detailed in Sec. 2.3.3.

To avoid confusion, we have added the following to the Fig. 5 caption: "The $O_3$ concentration data used as input for CLM5 are sourced from the ECMWF Atmospheric Composition Reanalysis 4 (EAC4)."

2.  Figures 3 and 4 set the foundation of the new scheme, as mentioned by Reviewer #1, the spread of the data point suggesting POD may not be the dominant factor driving the photosynthetic response. For instance, the fitted line for crop indicates a large reduction in photosynthetic rate with POD increase from 0 to 10 mmol m-2. Considering the large spread of the data points within this POD range, the concern would be this fitting line could be associated with large uncertainties. This uncertainty can propagate in the model and affect the O3 impact GPP, An, and gs analysis, such as the large reduction of An and gs of the crop in response to O3 in Figure 8.
**Reply:** Extensive field experiment analyses have identified POD as the best ozone index to link $O_3$-induced photosynthetic and stomatal responses so far, thus making it the most widely used index within both the observation and modeling communities. POD accounts for the cumulative $O_3$ uptake of plants, considering the comprehensive impact of $O_3$ concentration, exposure duration, and stomatal conductance (Explained in Para. 5 in the Introduction). Even though Wu et al. (2021) proposed a detoxification-capacity-based $O_3$ index that is better correlated to crop yield response based on observations, its required calculation of the difference between $O_3$ fluxes reaching cell wall surface and plasmalemma is currently an unfeasible task in existing global process-based land surface models.

The spread of data points in Figs. 3 and 4 arise from our aggregation of observations from various plant species into broader vegetation types, following the building of earlier schemes. Compared to the pursuit of explaining more variation of small data, we think a function that can represent the statistically significant $O_3$ response relationship observed in big data is more important and more suitable for global models that generalize global vegetation into several PFTs or biomes (rather than plant species). It's worth noting that all our response functions captured the observed statistically significant relationship between POD and photosynthetic /stomatal response. Introducing more explanatory variables (e.g., leaf trait variables as suggested by earlier studies) and conducting PFT-specific fitting may explain the variation better. These are the future development directions as listed in Sec. 5.1.4.

The larger reduction in the photosynthetic rate when POD is smaller (e.g., "0 to

10 mmol m-2 for crops") depicted by our fitting lines reflects the change in plant adaptability or transition from sensitivity to tolerance among plant species within a vegetation category, as observed in the real world and discussed in Sec. 5.1.2. Furthermore, our results that crops are sensitive to $O_3$ ("large reduction of An and gs of the crop in response to $O_3$ in Fig. 8") are consistent with earlier observational analyses (Reich 1987; Wang et al., 2024) and modeling works (Ma et al., 2023). We have added the comparison in Sec. 5.2.

We agree that our fitting lines could be associated with uncertainties that can propagate in the model and lead to uncertainty in quantifying the influence of ozone plant damage. We have added discussions on potential uncertainties and limitations of the parameterization scheme in Sec. 5.1.4 as follows: "Even though the new scheme has advantages over earlier schemes, as listed in the previous section, there are still noticeable variations in observations that have not been explained (Figs. 3 and 4). This limitation may introduce uncertainty in modeling carbon and water cycles, yield, biomass, and ecosystem structure and composition in large-scale process-based models, as well as in quantifying the role of ozone plant damage in the Earth system using these models to conduct numerical experiments.". In addition, the limitations of observations and suggestions to the observational community were presented in Sec.5.3.

Specific Remarks
1. BT and NT are defined in Line 336, but used before this line.
**Reply:** Thanks for pointing this out. In P7, we have added the definitions "BT and NT represent broadleaf trees and needle trees, respectively" in the title of Table 1.

2. Line 449-450: The manuscript states that the global reduction in leaf photosynthetic rate and stomatal conductance is 8.5% and 7.4%, respectively. It would be useful to provide a comparison with previous estimates from earlier schemes, either at global or regional scales, if any.
**Reply:** Thanks for your suggestion. We have added these comparisons to Sec. 5.2, along with all other comparisons: "Our estimates of the $O_3$-induced reduction in global average photosynthetic rate and stomatal conductance are around half of those calculated using the L15 (20.4% and 13.9%, Fig. 7). They are also lower than those estimated by Lombardozzi et al. (2013) (21% and 11%), which were derived from the average differences between control and $O_3$-fumigation experiments. Lombardozzi et al. (2013) used a smaller dataset than ours, did not differentiate between vegetation types or control experiment types, and did not filter out low-confidence data.".